# Loliolide, a New Therapeutic Option for Neurological Diseases? In Vitro Neuroprotective and Anti-Inflammatory Activities of a Monoterpenoid Lactone Isolated from *Codium tomentosum*

**DOI:** 10.3390/ijms22041888

**Published:** 2021-02-14

**Authors:** Joana Silva, Celso Alves, Alice Martins, Patrícia Susano, Marco Simões, Miguel Guedes, Stephanie Rehfeldt, Susete Pinteus, Helena Gaspar, Américo Rodrigues, Márcia Ines Goettert, Amparo Alfonso, Rui Pedrosa

**Affiliations:** 1MARE—Marine and Environmental Sciences Centre, Polytechnic of Leiria, 2520-630 Peniche, Portugal; celso.alves@ipleiria.pt (C.A.); alice.martins@ipleiria.pt (A.M.); patricia_susano94@hotmail.com (P.S.); marco.a.simoes@ipleiria.pt (M.S.); mg.cross@hotmail.com (M.G.); susete.pinteus@ipleiria.pt (S.P.); arodrigues@ipleiria.pt (A.R.); 2Department of Pharmacology, Faculty of Veterinary, University of Santiago de Compostela, 27002 Lugo, Spain; amparo.alfonso@usc.es; 3Cell Culture Laboratory, Graduate Program in Biotechnology, University of Vale do Taquari (Univates), Lajeado, RS 95914-014, Brazil; rehfeldt.stephanie@gmail.com (S.R.); marcia.goettert@univates.br (M.I.G.); 4BioISI—Biosystems and Integrative Sciences Institute, Faculty of Sciences, University of Lisbon, 1749-016 Lisboa, Portugal; hmgaspar@fc.ul.pt; 5MARE—Marine and Environmental Sciences Centre, ESTM, Polytechnic of Leiria, 2520-614 Peniche, Portugal

**Keywords:** antioxidant activity, inflammation, marine natural products, neuroprotection, NF-kB pathway, oxidative stress, Parkinson’s disease, seaweeds

## Abstract

Parkinsons Disease (PD) is the second most common neurodegenerative disease worldwide, and is characterized by a progressive degeneration of dopaminergic neurons. Without an effective treatment, it is crucial to find new therapeutic options to fight the neurodegenerative process, which may arise from marine resources. Accordingly, the goal of the present work was to evaluate the ability of the monoterpenoid lactone Loliolide, isolated from the green seaweed *Codium tomentosum*, to prevent neurological cell death mediated by the neurotoxin 6-hydroxydopamine (6-OHDA) on SH-SY5Y cells and their anti-inflammatory effects in RAW 264.7 macrophages. Loliolide was obtained from the diethyl ether extract, purified through column chromatography and identified by NMR spectroscopy. The neuroprotective effects were evaluated by the MTT method. Cells’ exposure to 6-OHDA in the presence of Loliolide led to an increase of cells’ viability in 40%, and this effect was mediated by mitochondrial protection, reduction of oxidative stress condition and apoptosis, and inhibition of the NF-kB pathway. Additionally, Loliolide also suppressed nitric oxide production and inhibited the production of TNF-α and IL-6 pro-inflammatory cytokines. The results suggest that Loliolide can inspire the development of new neuroprotective therapeutic agents and thus, more detailed studies should be considered to validate its pharmacological potential.

## 1. Introduction

Parkinson´s Disease (PD) is a chronic and progressive disorder of the central nervous system that affects about 6 million people worldwide. It is the most common neurodegenerative disease in aged people [1]. To date, the existent therapies only succeeded in relieving the clinical symptoms, but do not havecapacity to effectively deter the disease progression. Therefore, it is important to find new therapeutic agents that act not only to relieve symptoms, but also to slow down/block the disease progression [2]. Although the etiology of PD remains unclear, the death of dopaminergic (DA) neurons during PD progression is known to be associated with oxidative stress, mitochondrial dysfunction, and neuroinflammation [3,4]. Oxidative stress is the resulting condition of an imbalance between antioxidant defenses and the generation of oxidative species. Although numerous factors contribute for an increase of oxidative species production, the generation of reactive oxygen species (ROS) due to mitochondrial dysfunction is thought to be a main trigger of oxidative stress. If ROS production is not managed, these reactive species lead to a cascade of events resulting in neuronal cell death [4,5,6,7]. On the other hand, in PD is also observed an increase of the dopamine (DA) and nitric oxide (NO) metabolism, which, allied to reduced levels of endogenous antioxidant enzymes such as Catalase (CAT), glutathione peroxidase (GPx) and superoxide dismutase (SOD), also result in neuronal cell death [8,9]. Dopamine is an unstable molecule that undergoes auto-oxidation originating dopamine quinones and free radicals. These are metabolized by both A and B forms of monoamine oxidase (MAO), regulated through the oxidative metabolism of MAO-A that, with aging and neurological disorders increases MAO-B levels, becoming the predominant enzyme involved in dopamine metabolization [10]. One of the products of dopamine -MAO-B mediated metabolism is H_2_O_2_ that reacts with Fe(II) originating hydroxyl radicals (^•^OH), which target nigral dopaminergic neurons, promoting their loss [4]. Other sources of free radicals include the NO metabolism, which, when overexpressed, dysregulate the complexes I and IV of the mitochondrial electron transport chain, prompting ROS generation [11].

In an attempt to manage oxidative stress damage, several secondary defensive pathways are activated, including the release of inflammatory mediators such as the cytokines TNF-α, IL-1β, IL-6, resulting in inflammation [12,13]. Although inflammation is a protective strategy to promote tissue repair, chronical inflammation results in cellular damage, and when in the brain, neuronal cell loss, being also considered one of the main triggers of PD development [14,15]. On the other hand, there also occurs the release of the anti-inflammatory cytokine IL-10, which plays a critical role in the balance of immune responses, and thus can be a valuable biomarker in PD investigation [16]. In fact, a previous study reported the ability if IL-10 to decrease the number of activated microglia mediating a protective effect against the loss of dopaminergic neurons in the brains of a LPS-induced PD mouse model [17]. The increase of peripheral concentrations of IL-6, IL-1β, TNF-α, IL-2 and IL-10 cytokines was already observed PD patients [18].

The nuclear factor NF-kB is a protein complex that regulates cytokines production, playing a crucial role in the regulation of inflammation and apoptosis involved in the brain programming of systemic aging, as well as in the pathogenesis of several neurodegenerative diseases, including PD [19]. In an oxidative condition, NF-kB initiates the transcription of proinflammatory genes coding cytokines and proteolytic enzymes. However, NF-kB factor is composed by different dimers that can have either protective or noxious effects. For example, p50/RelA dimers induce pro-apoptotic and c-Rel-containing dimers exert neuroprotective actions [19]. Studies in another PD model also reported dopaminergic neuron loss in a 1-methyl-4-phenyl-1,2,3,6-tetrahydropyridine (MPTP)-mouse model suggesting that RelA upregulation may play a role in dopaminergic neuron degeneration when RelA is subexpressed [20]. On the other hand, studies in deficient mice for c-Rel subunit verified that this subunit can exert pro-survival effects, suggesting that a reduction in the protective function of c-Rel may render dopaminergic neurons more vulnerable to aging [19]. As a result, current research has been focused in finding substances that act to prevent mitochondrial dysfunction, oxidative stress, and inflammation for the development of more efficient therapies.

With increasing evidence of marine-derived molecules’ pharmacological effects, it is extremely relevant to understand their potential to be explored as novel PD therapeutics. Twenty-nine compounds isolated from marine organisms such as bacteria, fungi, seaweeds, sponges, corals, mollusks, sea cucumber and conus already demonstrated potential for PD therapeutics; however, only five compounds have entered in clinical trials [3]. Among marine organisms, seaweeds have shown to be relevant producers of bioactive compounds with neuroprotective potential [21,22,23] on several in vitro and in vivo PD models, including α-synuclein-, MPTP-, and 6-OHDA, revealing capacity to inhibit apoptosis, mitochondrial abnormalities and ROS production [24,25,26].

*Codium tomentosum* Stackhouse (Chlorophyta) is a green seaweed belonging to the Codiaceae family. This seaweed is native from the North East Atlantic Ocean, from British Isles southwards to Azores and Cape Verde and can be found on exposed rocks, and in deep rock pools on the lower shore [27]. Although it is a common seaweed on several coastlines all over the world, the existing studies with *C. tomentosum* metabolites are limited to fatty acids, organic acids, phenolics, and volatiles composition [28], and its bioactive potential mainly screened on crude extracts, including antioxidant, antigenotoxic, antimicrobial, antitumorigenic, neuroprotective, and hypoglycemic studies [22,29,30,31,32]. Recently, we studied the neuroprotective effect of the *C. tomentosum* fractions, which revealed capacity to recover the 6-OHDA-induced neurotoxicity, decreasing ROS production, mitochondrial dysfunction, and Caspase-3 activity [22]. Accordingly, the study presented here aims to isolate the compounds responsible for the activities mediated by *C. tomentosum* fractions. In the genus *Codium* sp. were already identified several sterols namely, decortinol, sodecortinol, decortinone, clerosterol and 3-*O*-β-D-galacto- pyranosyl clerosterol [33]. However, according to our knowledge, this study reveals for the first time the presence of Loliolide in *C. tomentosum* species.

Loliolide is an ubiquitous monoterpenoid lactone, firstly described in 1974, and isolated from plants and animals [34]. It was also found in marine ecosystems in different brown seaweeds, such as *Sargassum horneri* [35], *Sargassum ringgoldianum* subsp. *coreanum* [36], and *Undaria pinnatifida* [37], revealing antioxidant, anti-fungal, antibacterial and anti-cancer properties [34,35,36].

The present work is focused on the neuroprotective and anti-inflammatory potential of Loliolide obtained from *C. tomentosum*, and its underlying mechanisms of action, opening alternative research possibilities for the development of new PD therapeutic agents.

## 2. Results

### 2.1. Bioguided Fractionation of Codium tomentosum Extracts

#### 2.1.1. Extraction and Fractionation of *Codium tomentosum* Seaweed

*Codium tomentosum* was subjected to several fractionation steps resulting in three fractions (S1–S3) as illustrated in Figure 1.

#### 2.1.2. Antioxidant Activity of *Codium tomentosum* Fractions

The extraction yields, as well as the antioxidant capacity of each fraction (S1–S3) assessed through 2,2-diphenyl-1-picrylhydrazyl radical (DPPH) scavenging activity and ferric reducing antioxidant power (FRAP) assays, are summarized in Table 1.

As shown in Table 1, the highest extraction yields were achieved with water (S3—28.98%), while lower yields were obtained with ethyl acetate (S2—0.03%).

Concerning the antioxidant activities, none of the fractions demonstrated potential to reduce the DPPH radical. In the FRAP assay, S1 fraction revealed the highest efficiency to reduce ferric ions (82.01 ± 0.97 µM FeSO_4_/g extract), while the lowest activity was displayed by fractions S2 and S3 (8.54 ± 2.66 and 6.07 ± 0.16 µM FeSO_4_/g extract, respectively). However, when compared with synthetic antioxidant butylated hydroxytoluene (BHT), fraction S1 did not show a relevant antioxidant activity.

#### 2.1.3. Neuroprotective Potential of *Codium tomentosum* Fractions

The neuroprotective effects of *C. tomentosum* fractions (100 µg/mL; 24 h) were evaluated on neuroblastoma cell line (SH-SY5Y) exposed to the neurotoxin 6-OHDA (100 µM). Cell viability was estimated by the 3-(4,5-dimethyl-2-thiazolyl)-2,5-diphenyl-2H-tetrazolium bromide (MTT) assay, and the results are presented in Figure 2.

The exposure of SH-SY5Y cells to 6-OHDA (100 µM) for 24 h led to a reduction of cell viability of about 41% (59.69 ± 3.23% of viable cells) when compared to vehicle (100.00 ± 4.15% of viable cells). However, when SH-SY5Y cells were treated with 6-OHDA in the presence of *C. tomentosum* fractions (100 µg/mL), two samples exhibited capacity to prevent cell death in 15% and 40%, namely S3 and S1. The latter was selected for further purification processes.

### 2.2. Isolation and Structural Elucidation of Codium tomentosum Bioactive Compounds

Since fraction S1 attenuated the neurotoxicity induced by 6-OHDA, it was sub-fractionated by column chromatography aiming the isolation of the bioactive compound(s) having attained 10 sub-fractions (Figure 1). Sub-fraction F6 afforded the monoterpene lactone Loliolide, the structure of which was established by nuclear magnetic resonance (NMR) spectroscopy. The obtained ^13^C and ^1^H chemical shifts and structural assignments (Table 2) are in accordance with the literature [38].

### 2.3. Antioxidant Capacity of Loliolide

The antioxidant activity of Loliolide was evaluated by three different approaches, namely, DPPH radical scavenging ability, oxygen radical absorbance capacity (ORAC), and ferric reducing antioxidant power (FRAP) methods. It was found that Loliolide did not present a noticeable DPPH radical scavenging ability (EC_50_ > 100 µM). Additionally, in the ORAC (24.22 ± 3.45 µmol TE/g) and FRAP (13.81 ± 1.36 µM FeSO_4_/g) assays, this compound also has not shown capacity for reducing peroxyl radicals and ferric ions, when compared with BHT (143.70 ± 23.36 µmol TE/g and 2821.50 ± 63.03 µM FeSO_4_/g, respectively).

### 2.4. Bioactivity Evaluation of Loliolide on In Vitro Cellular Models

#### 2.4.1. Cell Viability and Neuroprotective Effects of Loliolide on SH-SY5Y Cells

In a first approach, the possibility of Loliolide to induce cytotoxic effects on SH-SY5Y cells was considered (Figure 3A). Subsequently, the neuroprotective effects of Loliolide (1, 5, 10, 50, 100 µM; 24 h) were evaluated in the presence of the 6-OHDA neurotoxin (Figure 3B).

The results suggest that Loliolide did not present cytotoxicity and exhibited high neuroprotective effects against the 6-OHDA neurotoxin, leading to an increase of cell viability of 23.70 ± 7.77% and 41.06 ± 6.31%, at 50 µM and 100 µM, respectively.

#### 2.4.2. Effects of Loliolide on PD-Hallmarks

Several PD hallmarks associated with Loliolide neuroprotective effects on SH-SY5Y cells were evaluated, namely ROS production, Catalase activity, mitochondrial membrane potential (MMP), adenosine triphosphate (ATP) levels and Caspase-3 activity. These hallmarks were evaluated on SH-SY5Y cells treated with 6-OHDA in the presence and absence of Loliolide (50–100 µM), and results are presented in Figure 4.

In all biomarkers, the effect of Loliolide in the absence of 6-OHDA was also evaluated, and no significant differences were observed when compared to vehicle.

The exposure of cells to 6-OHDA (100 µM) lead to a marked increase of ROS levels when compared to vehicle (Figure 4A). In the presence of Loliolide, both concentrations induced a significant decrease of ROS levels. The Catalase activity, a stress condition biomarker, was also evaluated representing a stequiometry of 1:1 with H_2_O_2_ levels. The exposure to 6-OHDA led to an increase of Catalase activity of 24.00 ± 1.03%, when compared with vehicle (Figure 4B). When in the presence of Loliolide, Catalase activity increased to 60.40 ± 3.72% at 100 µM. To understand if the neuroprotective effects of Loliolide were mediated by mitochondrial events, the MMP and ATP levels were evaluated. In the cells exposed to 6-OHDA it was verified a depolarization of the MMP of 206.80 ± 24.55%, accompanied by a 23.36 ± 1.49% decrease of ATP levels, when compared with vehicle (Figure 4C,D). The treatment with Loliolide prevented MMP distress; however, it did not stabilize the ATP levels disrupted by 6-OHDA. Caspase–3 activity was determined to understand if the neuroprotective potential of Loliolide was mediated by apoptotic pathway. Figure 4E shows that when SH-SY5Y cells are exposed to 6-OHDA, Caspase–3 activity is increased by about 200% when compared with the vehicle. On the other hand, the treatment with Loliolide promoted a decrease of Caspase–3 activity of 110.1 ± 37.82% and 126.1 ± 48.95% at 50 µM and 100 µM, respectively.

#### 2.4.3. DNA Fragmentation and Nuclear Condensation

To understand if Loliolide had the ability to prevent the DNA damage and nuclear condensation induced by 6-OHDA treatment, the integrity of SH-SY5Y cells DNA was evaluated through the staining with DAPI probe, and the results are presented in Figure 5.

Figure 5 shows nuclear fragmentation and condensation of SH-SY5Y cells resulting from the exposure to 6-OHDA. The treatment with Loliolide (50 and 100 µM; 24 h) inhibited nuclear condensation and DNA fragmentation.

#### 2.4.4. Loliolide Effects on NF-kB p65 Translocation

Nuclear transcription factor-kB (NF-kB) is an apoptosis/inflammation regulator present in human tissues, including brain. Thus, we tested if Loliolide affected the translocation of this protein to the nucleus of neuronal cells. The expression levels of NF-kB p65 in cytosol and nucleus were studied by Western blot (Figure 6).

As shown in Figure 6A, the neurotoxicity induced by 6-OHDA led to an increase of NF-κB p65 expression in the nucleus accompanied by a decrease in the cytoplasm, when compared with the vehicle (1.39 ± 0.19 and 1.53 ± 0.25, respectively). However, when cells were treated with 6-OHDA in the presence of Loliolide, the nuclear NF-kB p65 levels were downregulated and cytoplasmic NF-kB p65 levels were upregulated. However, at 100 µM, Loliolide exhibited significant ability to block the translocation of NF-κB p65 factor from the cytoplasm to the nucleus, being also possible to observe an increased expression levels of NF-κB p65 factor, when compared with the nucleus levels. On the other hand, in the treatment performed only with 6-OHDA an opposite effect was observed, suggesting that NF-κB p65 factor translocation inhibition may be associated with protective effects of Loliolide against neuronal cell death mediated by 6-OHDA.

### 2.5. Anti-Inflammatory Activity of Loliolide on RAW 264.7 Cells

In a first approach, Loliolide was tested for possible cytotoxic and inflammatory effects on murine macrophages (RAW 264.7 cells). The anti-inflammatory potential of Loliolide (50–100 µM; 24 h) was then determined on RAW 264.7 cells, in an inflammatory condition mediated by lipopolysaccharides (LPS). The results are presented in Figure 7.

In Figure 7, it is possible to observe that Loliolide did not induce cytotoxicity on RAW 264.7 cells at 50 and 100 µM (Figure 7A). Furthermore, the treatment of RAW 264.7 cells with Loliolide at 50 µM and 100 µM did not stimulate the NO production (Figure 7B) when compared with the vehicle. On the other hand, the exposure to LPS stimulated the NO production. However, when LPS-stimulated RAW 264.7 cells were treated with Loliolide at 50 µM and 100 µM the NO production decreased significantly (Figure 7C) when compared with LPS situation.

### 2.6. Effects of Loliolide on the Pro-Inflammatory and Anti-Inflammatory Cytokines Levels

RAW 264.7 cells were exposed to LPS and Loliolide, and the levels of the pro-inflammatory cytokines TNF-α, IL-6, and anti-inflammatory IL-10 were determined by ELISA. The results are depicted in Figure 8.

In response to LPS stimulation, the production of cytokines was significantly upregulated compared to the control, except IL-10. The treatment with Loliolide significantly reduced the production of TNF-α by up to 305% and 238% at 50 µM and 100 µM, respectively (Figure 8A). The treatment with Loliolide also significantly reduced the production of IL-6 by up to 197% and 220% at 50 µM and 100 µM, respectively (Figure 8B). However, it did not promote significant effects on the IL-10 levels (Figure 8C).

## 3. Discussion

PD is the second most common neurodegenerative disease worldwide being characterized by a progressive degeneration of nigrostriatal dopaminergic neurons [39,40]. Currently, the molecular mechanisms underlying the loss of these neurons still remain vague; however, mitochondrial dysfunction, apoptosis, and neuroinflammation are thought to play an important role in dopaminergic neurotoxicity in PD [4]. In all these biological events, oxidative stress is the common underlying mechanism that leads to cellular dysfunction and cell death. Despite the efforts, PD treatment continues to be a major clinical challenge, without an effective cure. Therefore, more studies are needed to understand the pathomechanisms of PD to find effective neuroprotective agent(s).

In the present study, the antioxidant, neuroprotective and anti-inflammatory activity of Loliolide isolated from *C. tomentosum* was evaluated. Regarding its antioxidant capacity, three methods (DPPH, FRAP and ORAC) were assayed but in none Loliolide showed significant antioxidant potential, which is in agreement with a previous work [41]. However, despite the weak antioxidant activity observed in chemical assays, when tested on an in vitro cellular model exposed to UVA-B radiation, this compound inhibited the ROS production and apoptosis [42].

The antioxidant activity of a compound should not be determined based on a unique in vitro test since those have the limitation of targeting a specific oxidative species (e.g., alkoxy and peroxyl radicals, reactive nitrogen species, ROS) [43,44], disregarding all cellular oxidative pathways players, and thus, may not reflect the antioxidant ability in more complex models or in vivo. Since several studies have associated dopaminergic cell death prevention with antioxidative mechanisms [45,46], in this work, the antioxidant potential of Loliolide was screened in a PD cellular model. Neurotoxicity was induced by 6-OHDA, a neurotoxic compound that selectively destroys dopaminergic and noradrenergic neurons in the brain by instigating ROS production [47].

Marine-derived compounds have been previously reported to exhibit neuroprotective effects against 6-OHDA neurotoxicity [23,48]. In this study, Loliolide revealed capacity to promote SH-SY5Y cells recover from the damage induced by 6-OHDA. The results presented herein are consistent with previous studies. For instance, 11-dehydrosinulariolide, a soft coral-derived compound, and Kappa-carrageenan, isolated from *Hypnea musciformis* seaweed, had capacity to protect SH-SY5Y cells against the neurotoxic effects of 6-OHDA [23,48]. Also, Yurchenko et al. (2018) [49] showed that metabolites isolated from the marine fungi *Penicillium* sp., *Aspergillus* sp. and *Aspergillus flocculosus* protected Neuro2a cells against the damaging effects of 6-OHDA. The death of dopaminergic neurons occurs in the central nervous system (CNS), where the neurotransmitter dopamine is synthesized and released into the substantia nigra [50]. Currently, it is known that 70% of those dopaminergic neurons in CNS die along PD development; however, the symptoms are not noticed due to compensatory mechanisms of the striatum body [51]. Although the mechanisms responsible for inducing cell death in PD are not completely clarified, it is known that the dopamine metabolism, oxidative stress, mitochondrial dysfunction and neuroinflammation are involved in the death of dopaminergic neurons in CNS [4,52]. Iron (Fe(II)) plays an important role in oxidative changes in PD, being present in several regions of the brain, mainly in dopaminergic neurons of the substantia nigra [53]. As a result, dopaminergic neurons are highly susceptible to Fenton’s reaction, in which H_2_O_2_ is converted to hydroxyl radicals promoting oxidative stress, leading to apoptosis, DNA damage and cellular death.

There many enzymes are able to decompose H_2_O_2_, including Catalase, glutathione peroxidase and other peroxidases [54]. Catalase is a key detoxifying enzyme that uses H_2_O_2_ as substrate, decomposing it into water and oxygen, maintaining an optimum level of this molecule in the cells [55]. Based on the results obtained in the present work, it was observed that Loliolide was able to decrease the ROS production induced by 6-OHDA treatment and stimulate Catalase activity. These results are in agreement with those previously reported by Magalingam and co-workers [56] that also observed a significant increase of Catalase activity in PC-12 cells exposed to 6-OHDA in the presence of antioxidants, namely rutin and isoquercitrin, when compared with 6-OHDA situation. Regarding 6-OHDA, it was observed a slight increase of Catalase activity that can be explained by the cellular metabolism response to the increase of ROS levels, such as H_2_O_2_, mediated by 6-OHDA treatment, leading to a stimulation of enzyme activity in order to detoxify H_2_O_2_. On the other hand, the weak antioxidant capacity of Loliolide and its ability to improve Catalase activity suggest that its activity may not be directly related with the neutralization of ROS but with the stimulation of the antioxidant defense machinery. This point of view is reinforced by the results attained with Loliolide when tested at 100 µM, and it is possible to observe that the highest increase of Catalase activity was accompanied by the highest decrease of ROS levels.

Oxidative stress has been intimately linked to mitochondrial dysfunction. Mitochondria are vitally important organelles involved in energy metabolism, generating over 90% of our cellular energy in the form of ATP, through oxidative phosphorylation, and they are involved in various other processes, including the regulation of calcium homeostasis, and stress response [57]. As a consequence of these processes, mitochondria are also responsible for more than 90% of cellular ROS production [42], and are completely dependent of an efficient antioxidant machinery to prevent oxidative stress. In clinical pathologies such as PD, mitochondrial dysfunction is a characteristic abnormality. An increase of ROS levels is detected, against which antioxidative defenses are overwhelmed [58]. Based on the results presented here, it was verified that Loliolide prevented mitochondrial dysfunction, decreased Caspase-3 activity and inhibited DNA fragmentation/condensation (apoptosis morphological trait) disrupted by 6-OHDA. However, it did not exhibit capacity to recover the ATP levels decreased by 6-OHDA treatment. Since Loliolide was able to decrease the mitochondrial depolarization induced by 6-OHDA exposure, we hypothesize that it would be necessary to increase the incubation time in order to be possible to observe a potential increase of ATP levels. Nevertheless, further studies should be considered to verify this hypothesis.

Several pieces of evidence show that the activation of the transcription factor NF-kB is associated with oxidative stress-induced apoptosis and some researchers verified that this transcription factor can play a role in PD when translocated from the cytoplasm to the nucleus [59]. Under normal conditions, NF-κB is bounded in the cytoplasm to the inhibitor protein, IκBα, which sequesters NF-κB in the cytosol, inactivating its transcription factor by masking the nuclear localization signals of NF-κB proteins. The activation of NF-kB involves its dissociation from IκBα followed by its translocation to nucleus, where it directly binds to DNA sequences, inducing damage [60]. The present results showed that Loliolide had capacity to inhibit NF-kB translocation from the cytoplasm to nucleus, thus preventing DNA damage and apoptosis. Our results are in agreement with Alvariño et al. [61], who reported that gracilin A, a marine-derivates have capacity to downregulate the NF-κB factor, prompted by H_2_O_2_ in microglia cells. Furthermore, due to the key role of NF-kB factor in the response to oxidative stress and inflammation, the effects mediated by Loliolide in the other biomarkers may be triggered through the translocation of this factor.

Chronic inflammation is characteristic of neurodegenerative diseases such as PD, in which cytotoxic levels of NO and pro-inflammatory cytokines initiate neuronal cell death pathways. Several studies reported that microglia activation could have a protective role in neurodegenerative diseases [13,14]. Therefore, promoting anti-inflammatory cytokines or limiting pro-inflammatory cytokines and NO production by macrophages/microglia activation, can be beneficial for preventing inflammation [13]. The macrophages are known to cross the leaky blood-brain barrier in PD to interact with microglia and stimulate the secretion of inflammatory cytokines causing brain damage via neuroinflammation. Thus, the uncontrolled release of inflammatory cytokines, such as TNF-α and IL-6, is a key event for neurodegenerative diseases progression [62,63]. As a result, the proximity of microglia with macrophages attracted increasing attention in relation to the onset and PD progression. In this study, macrophages were exposed to LPS to induce inflammation [64] in the presence of Loliolide. It was verified that Loliolide prevented inflammation by decreasing NO production and reducing TNF-α and IL-6 levels. The interleukin-10 (IL-10) is a potent anti-inflammatory cytokine that plays a vital role in immunologic system, acting in acute and chronic inflammation. Both up or downregulation of IL-10 cytokine will result in serious immunologic disorders, including neurodegenerative diseases. Studies performed with the brains of a PD mouse model induced with LPS evidenced that IL-10 cytokine mediated a decrease of the number of activated microglia promoting a protective effect regarding the loss of dopaminergic neurons [16]. In the present work, the treatment of RAW 264.7 cells with LPS, in the presence/absence of Loliolide, did not induced significant changes on IL-10 levels. Although the IL-10 generally mediated effects are able to oppose the actions mediated by the pro-inflammatory cytokines, its activity is highly complex in the immunoregulation that refines IL-10 production to a later stage when compared to pro-inflammatory cytokines. Thus, it is possible that it would be necessary to prolong cell exposure to both the inducer (LPS) and the protective agent (Loliolide) to detect this protein production [65]. Accordingly, with the above-mentioned facts, and knowing that p38 MAPK pathway plays a central role in the inflammation process, stimulating the release of pro-inflammatory cytokines and activating NF-kB transcription factor, it is possible that Loliolide effects may be also linked to p38 activity inhibition. However, additional studies need to be accomplished to prove this hypothesis.

More recently, there are studies indicating that the development of chronic neuroinflammatory diseases can be influenced by intestinal inflammation. Gut bacteria can release factors and metabolites into the blood that can readily cross blood–brain barrier (BBB) or otherwise interact with barrier cells, changing its integrity, and transport rates, thus affecting CNS regulation and functions [66]. Furthermore, metabolic products synthesized by the microbiota when crossing the BBB trigger the inflammatory cascade, including microglial activation and neuronal dysfunction leading to the death of neurons. More recently, previous studies conducted with PD patients have shown evidence of an altered intestinal microbiota, systemically releasing endotoxins such as lipopolysaccharides and metabolic products facilitating their entry into the CNS promoting the activation of the microglia inducing inflammatory responses, such as the release of pro-inflammatory cytokines (IL1-α, IL-β, IL-6 and TNF-α), leading to the degeneration of dopaminergic neurons [66,67]. Therefore, due to the anti-inflammatory properties exhibited by Loliolide, as well as its ability to cross the BBB as described by Ahmed et al. [68], its application as a dietary anti-inflammatory molecule can represent an excellent strategy to contribute for PD therapeutics.

Loliolide showed capacity to protect neuronal cells from the damaging effects of 6-OHDA, neutralizing its cytotoxicity, preventing mitochondrial dysfunction, apoptosis and also showing high anti-inflammatory effects, inhibiting the NF-kB pathway, decreasing the levels of TNF-α and IL-6, thus blocking the inflammatory cascade.

Due to promising activities exhibited by Loliolide in the present work, further studies should be considered in order to fully depict the intracellular signaling pathways underlying its neuroprotective and anti-inflammatory activities, including the study of protein expression levels of other endogenous antioxidant enzymes (e.g., SOD, GSH-Px), other apoptosis biomarkers (e.g., Caspases -9 and -8, Bax, cytochrome C) as well as the study of others factors/proteins related with antioxidant, neuroprotective and anti-inflammatory biological processes (e.g., COX, iNOS, PGE2, JAK, JNK, Nrf2/HO-1, α-synuclein expression). Furthermore, the establishment of more complex in vitro cellular models, such as co-cultures and 3D cell models, as well as the development of smart delivery systems such as nanoparticles for brain drug delivery will be essential to understand the really therapeutic potential of Loliolide on PD.

In conclusion, the monoterpene lactone Loliolide, isolated from the green algae *C. tomentosum*, presented capacity to protect neuronal cells from the damaging effects of 6-OHDA, neutralizing its cytotoxicity. Additionally, the studies here performed suggest that this compound acts in several PD hallmarks, reducing oxidative stress, preventing mitochondrial dysfunction, and blocking inflammatory pathways, thus preventing neuronal cell death (Figure 9). This work highlights Loliolide as a promising compound for PD therapeutics and, therefore, should be considered for additional detailed studies in more complex models, aiming at further considerations for pre-clinical trials.

## 4. Materials and Methods

### 4.1. Collection and Preparation of Codium tomentosum Samples

The green seaweed *C. tomentum* Stackouse, (1797) was collected in October 2016 at Peniche coast, Portugal (39°37′05″ N, −9°38′58″ O) and transported to MARE-Polytechnic of Leiria lab facilities. The seaweed was rinsed carefully with seawater and distilled water to remove epiphytes, sand, and debris. Then, it was freeze-dried (Scanvac Cool Safe, LaboGene, Lynge, Denmark), grinded, and stored in a cool place protected from light, until further use.

### 4.2. Seaweed Extraction and Fractionation

The freeze-dried biomass of *C. tomentosum* (445.0 g) was extracted with methanol (VWR-BDH Chemicals, Fontenay-sous-Bois, France) (in a biomass/solvent ratio of 1:40) overnight, under constant stirring at room temperature. The methanol extract was concentrated until dryness under vacuum, at low temperature (30 °C) in a rotary evaporator (IKA HB10, Staufen, Germany) and in a speed vacuum equipment (Concentrator Plus, Eppendorf, Spain), affording the MeOH crude extract (112.3 g). Afterwards, it was suspended in hot (80 °C) water (400 mL) and filtered (filter paper no. 4, VWR International, Alfragide, Portugal). After cooling to room temperature, the aqueous phase was partitioned, firstly with diethyl ether (5 × 200 mL), and then with ethyl acetate (3 × 200 mL). Organic phases were dried with anhydrous Na_2_SO_4_, filtered (filter paper no. 4), and concentrated to dryness, resulting in three fractions (S1–S3) (Figure 1).

The dried diethyl ether extract (346.0 mg) (S1) was fractionated by preparative column chromatography on silica gel 60 (0.06–0.2 mm, VWR, Leuven, Belgium) eluted with mixtures of *n*-hexane, ethyl acetate, and methanol (Fisher Scientific, Loughborough, UK) of increasing polarity, affording a total of 10 fractions. Fractions were screened by thin layer chromatography in TLC-plates ALUGRAM^®^ Xtra Sil G/UV254, pre-coated with silica gel 60 (Merck, Lisbon, Portugal), and those exhibiting the same chemical profile were pooled. Fraction 6 (Hex: EA, 60:40, *v/v*) yielded the bioactive compound **1** (5.0 mg) (Figure 1), the structure of which was attained by NMR spectroscopy.

### 4.3. Structural Elucidation of the Bioactive Compound

NMR spectra of 1D (^1^H, ^13^C APT) and 2D (COSY, HMBC, HSQC and NOESY) experiments were acquired on a Bruker Advance 400 spectrometer with a frequency of 400 MHz for ^1^H, and 100 MHz for ^13^C. Compound **1** was dissolved in 500 µL of CDCl_3_ (Sigma-Aldrich, St. Louis, MO, USA). Chemical shifts are expressed in ppm and reported to the residual solvent signals. Coupling constants (*J*) are expressed in Hertz (Hz).

### 4.4. Bioactivity Assays

Fractions S1–S3 and compound **1**, from *C. tomentosum*, were subjected to a series of in vitro biological assays, to evaluate their antioxidant, neuroprotective, and anti-inflammatory potential.

#### 4.4.1. Antioxidant Capacity

The antioxidant activity was evaluated according to Silva et al. [21] using different approaches, namely, 2,2-diphenyl-1-picrylhydrazyl (DPPH) radical scavenging ability, oxygen radical absorbance capacity (ORAC), and ferric reducing antioxidant power (FRAP) assays.

#### 4.4.2. Cell Culture Maintenance

Neuroblastoma cell line (SH-SY5Y) was obtained from the German Collection of Microorganisms and Cell Cultures GmbH (DSMZ) Bank (ACC 209). Cells were cultivated at 37 °C and 5% CO_2_ with DMEM:F12 medium containing 1% antibiotic/antimycotic (Amphotericin B, Penicillin and Streptomycin) (Biowest, Nuaillé, France) and 10% (*v/v*) Fetal Bovine Serum (FBS) (Biowest, Riverside, MO, USA). Cells were seeded as follows: 4 × 10^4^ cells/well, in 96-well microplates for cell viability and neuroprotection assays; 2 × 10^5^ cells in 12-well culture plates for Catalase assay; 5 × 10^5^ cells in 6-well culture plates for Caspase–3 activity, nuclear morphological damage assessment, and NF-kB p65 translocation analysis.

#### 4.4.3. Cell Viability and Neuroprotective Capacity on a PD Cellular Model

Effects on cell viability and neuroprotection assays were estimated using the MTT (VWR, Solon, Ohio, USA) method, as described by Silva et al. [22]. SH-SY5Y cells were exposed 24 h to 6-OHDA (100 µM) in absence (Vehicle) or presence of compound **1** at concentrations of 1–100 μM. Then, 100 µL MTT (1.2 mM) were added to wells and cells were incubated for 1 h at 37 °C. After this time, MTT was removed and 100 µL DMSO was added. The resulting absorbance was read in a microplate reader (Bioteck, Epoch/2 microplate reader, Winooski, VT, USA) at 570 nm. The results were expressed in percentage of control.

#### 4.4.4. Parkinson’s Disease Hallmarks

ROS production: ROS levels were evaluated using the 5(6)-carboxy-2′, 7′-dichlorofluorescein diacetate (carboxy-H2DCFDA) probe (Invitrogen, Bleiswijk, Netherlands) according to Alvariño et al. [69] with slight modifications. After incubation with compound **1,** cells were washed with ice-cold Phosphate-Buffer Saline (PBS) and C-H2DCFDA (100 µL, 20 µM) was added, and incubated for 1 h at 37 °C. The resulting fluorescence was read at 527 nm excitation and 590 nm emission wavelengths, and ROS levels presented in percentage of control (non-treated cells).

Catalase activity: This assay was assessed using the “Amplex ™ Red Catalase assay” kit (Invitrogen, Renfrew, UK). In this assay, catalase first reacts with 40 µM H_2_O_2_ to produce water and oxygen (O_2_). Next, the Amplex Red reagent reacts (1:1 stoichiometry) with any unreacted H_2_O_2_ in the presence of horseradish peroxidase (HRP), producing a highly fluorescent oxidation product, resorufin. Therefore, the signal from resorufin decreases as catalase activity increases.

SH-SY5Y cells were cultured in 12 -well plates and exposed 6 h to 6-OHDA (100 µM) in the absence (vehicle) and presence of compound **1** (50 and 100 µM). Cells were then rinsed with ice-cold PBS, followed by the addition of 100 µL of lysis buffer (20 mM Tris-HCl pH 7.4, 10 mM NaCl and 3 mM MgCl_2_), containing a protease inhibitor’ cocktail (Roche, Mannheim, Germany). Following, cells were scrapped, incubated on ice for 15 min and centrifuged at 1200× *g*, 4 °C for 10 min. Cells lysates were processed according to manufacturer’s instructions. H_2_O_2_ levels were determined in real time for 30 min at 37 °C. The fluorescence was read at 530 nm, with an excitation wavelength of 590 nm. Catalase activity was calculated by the slope of the linear phase of the fluorescence resulting from the resorufin oxidation, and the results were expressed in percentage of control.

Mitochondrial membrane potential (ΔΨm): MMP was evaluated according to Silva et al. [21] using JC-1 probe (Molecular Probes, Eugene, OR, USA). SH-SY5Y cells were exposed 6 h to 6-OHDA (100 μM) in the presence or absence of compound **1**. Cells were washed with ice-cold PBS, and then, 200 µL of JC-1 (3 µM) were added prior to 15 min at 37 °C. Afterward, JC-1 was removed, and PBS was added. The monomers/aggregates formation were determined by fluorescence extrapolation at 530 nm emission (monomers)/590 nm (aggregates) and 490 nm excitation wavelengths, for 30 min at 37 °C. MMP was calculated through the ratio between monomers/aggregates formation and presented as percentage of control.

Adenosine triphosphate (ATP) levels: The ATP levels were assessed using the “Luminescent ATP detection assay” kit (ABCAM, Cambridge, UK), according to the manufacturer’s instructions, and is based on the production of light derived from the reaction of ATP with luciferase and luciferin. The emitted light is proportional to the ATP concentration inside the cells.

SH-SY5Y cells were cultured in 96 -well plates and exposed 6 h to 6-OHDA (100 µM) in the presence or absence of compound **1** (50–100 µM). Later, a detergent solution (50 µL) was added and incubated for 5 min at room temperature, prior to the addition of 50 µL of a substrate solution. After 5 min incubation at room temperature, the ATP levels were measured by luminescence with a luminometer (BioTeck, Synergy H1 microplate reader, Winooski, VT, USA). Results were expressed as percentage of control.

Caspase-3 activity: Caspase-3 activity was evaluated according to Silva et al. [21]. The cells were rinsed with ice-cold PBS, scrapped, and centrifuged at 3300× *g*, for 5 min. After, SH-SY5Y cells were incubated on ice for 20 min with lysis buffer and finally centrifuged at 22,500× *g*, 4 °C, for 20 min. Cells lysates were processed following the manufacturer’s protocol “Caspase-3 fluorometric assay” (Sigma, St. Louis, MO, USA) and fluorescence was read at 360 nm excitation and 460 nm emission wavelengths. Caspase–3 activity was calculated through the curve slope and presented as percentage of control.

Nuclear morphological changes: 4′,6-diamidino-2-phenylindole (DAPI) (Applichem, Darmstadt, Germany) assay was conducted according to Lee et al. [70] with minor modifications [21] to determine nuclear morphological changes promoted by 6-OHDA. SH-SY5Y cells were cultured in 6-well plates and exposed 24 h to 6-OHDA (100 µM) in the presence or absence of compound **1** (50–100 µM). The cells were washed twice with ice-cold PBS and fixed with 4% paraformaldehyde (Fisher Scientific, Loughborough, UK) in PBS for 30 min. Fixed cells were washed with PBS and permeabilized with 0.1% Triton X-100 (Sigma, St. Louis, MO, USA) in PBS for 30 min. After washing again with PBS, the cells were stained with DAPI staining solution for 30 min at room temperature. The stained cells were observed under a fluorescence microscope (Zeiss, Axio Vert. A1, Oberkochen, Germany) to confirm the presence of apoptotic signs, such as size-reduced nuclei, chromatin condensation, intense fluorescence, and nuclear fragmentation.

Cytosolic and nuclear protein determination: SH-SY5Y cells were exposed to 6-OHDA (100 µM) in the presence or absence of compound **1** (50–100 µM) for 6 h. The protein content was determined according to Alvariño et al. [61], with slight modifications. Cells were rinsed with ice-cold PBS and a hypotonic buffer was added: 20 mM Tris-HCl pH 7.4 (Biorad, Hercules, CA, USA), 10 mM NaCl (Merck, Darmstadt, Germany) and 3 mM MgCl_2_ (Sigma, Buchs, Stwitzerland), containing a phosphatase and protease inhibitor cocktail. After the addition of the hypotonic buffer, the cells were incubated for 15 min on ice and centrifuged at 1200× *g*, for 15 min at 4 °C. The supernatant was collected (cytosolic fraction) and the pellet was resuspended in a nuclear extraction solution: 100 mM Tris pH 7.4, 2 mM Na_3_VO_4_ (Sigma, Karnataka, India), 100 mM NaCl, 1% Triton X-100 (Sigma, St. Louis, MO, USA), 1 mM EDTA (Sigma, Taufkirchen, Germany), 10% glycerol (Fisher Chemical, Geel, Belgium), 1 mM EGTA (Sigma, Buchs, Stwitzerland), 0.1% SDS (Biorad, Higashi-shinagawa Shinagawa-ku, Tokyo, Japan), 1 mM NaF (Fischer Chemical, Loughborough, UK), 0.5% deoxycholate (Sigma, Auckland, New Zealand), containing 1 mM phenylmethylsulfonyl fluoride (PMSF) (Sigma, Taufkirchen, Germany) and a protease inhibitor’ cocktail (Roche, Mannheim, Germany). Lysates were incubated for 30 min on ice, with vortexing intervals of 10 min. Then, samples were centrifuged at 14,000× *g*, at 4 °C for 30 min, and the nuclear fraction was obtained. Cytosolic and nuclear protein concentration was determined by the Lowry method according to Waterborg and Matthews [71] with some modifications.

NF-kB p65 translocation analysis: NF-kB p65 translocation analysis was conducted by Western blot as follows: electrophoresis was carried out in 10% sodium dodecyl sulfate polyacrylamide gels (Biorad, Hercules, CA, USA) with 20 or 10 µg of cytosolic or nuclear protein, respectively. After transferring the protein bands to polyvinylidene difluoride (PVDF) membranes (Biorad, Hercules, CA, USA) using the Trans-Blot Turbo transfer system (Biorad, Hercules, CA, USA), blocking was carried out with 5% skim milk (Panreac-Applichem, Darmstadt, Germany), tris-buffered saline (TBS) containing Tween-20 (Biorad, Hercules, CA, USA) for 1 h. Primary antibodies were added to the membranes and incubated overnight at 4 °C with continuous agitation. After this time, the membranes were washed with a mixture of tris-buffered saline with 0.1% Tween-20 (TBST), at room temperature, and the membranes incubated with their respective enzyme-linked secondary antibodies for 2 h at room temperature. The antibodies anti-NF-kB p65 (1:1000) (Santa Cruz Biotechnology, Dallas, TX, USA) were used for the evaluation of protein expression and the signal was normalized using anti-Lamin B1 (1:5000) (Santa Cruz Biotechnology, Dallas, TX, USA) and anti-β-actin (1:5000) (Santa Cruz Biotechnology, Dallas, TX, USA) for nuclear and cytosolic fractions, respectively. Protein bands were detected with “SuperSignal ^TM^ West Pico Plus Chemiluminescent Substrate” (Thermo Scientific Inc., Rockford, IL, USA) on a Chemidoc ^TM^ MP imaging System (Biorad, Hercules, CA, USA).

### 4.5. Anti-Inflammatory Proprieties on RAW 264.7 Cells

Cell culture maintenance: Murine macrophage cells (RAW 264.7) were obtained from ATCC Bank (TIB-71). Cells were cultured at 37 °C with 5% CO_2_ on DMEM medium without phenol red (Sigma, St. Louis, MO, USA), containing 1% antimycotic, 10% (*v/v*) fetal bovine serum and 1% pyruvate sodium. The cells were seeded with 5 × 10^4^ cells/well in 96-well microplates for cell viability and NO production determination, and with 5 × 10^5^ cells/well in 12-well plates for interleukins levels determination.

#### 4.5.1. Cell Viability and Nitric Oxide Production on Lipopolysaccharide (LPS) Inflammation Model

In this assay, RAW 264.7 cells were subjected to an inflammatory condition mediated by lipopolysaccharides (LPS) [72]. The inflammation status was estimated by the quantification of NO levels, which are directly proportional to inflammation. RAW 264.7 cells were pretreated with compound **1** (50–100 μM) for 1 h, and stimulated with 1 µg/mL LPS for 24 h. Cells’ viability was then determined by the MTT assay as previously described [73] and NO production determined using the Griess reagent (1% sulphanilamide (Alfa Aesar, Karlsruhe, Germany) in phosphoric acid (2.5%) (Merck, Darmstadt, Germany) with 0.1% naphtylethylenediamine dihydrochloride (Alfa Aesar, Ward Hill, MA, USA). The results are expressed in percentage of control.

#### 4.5.2. Measurement of Proinflammatory and Anti-inflammatory Cytokines Production

RAW 264.7 cells (5 × 10^5^ cells/mL) were pre-incubated for 1 h with compound **1** (50–100 µM) prior to incubation with LPS (1 µg/mL) at 37 °C for 18 h. The concentration of TNF-α, IL-6 and IL-10 cytokines (Thermoscientific, Vienna, Austria) were assayed using enzyme-linked immunosorbent assay (ELISA) kits according to the manufacturer’s instructions.

### 4.6. Data and Statistical Analysis

The results are presented as mean ± standard error of the mean (SEM). The determination of EC_50_ was attained from sigmoidal dose–response variable-slope curves using the GraphPad Prism V.8 software (GraphPad Software Inc., San Diego, CA, USA). One-way analysis of variance (ANOVA) with Dunnett’s multiple comparison of group means was employed to determine significant differences relatively to the control treatment. All data were checked for normality (Shapiro-Wilk test) and homoscedasticity (Levene’s test). Comparisons concerning variables, which did not meet variance or distributional assumptions, were carried out with Kruskal–Wallis non-parametric tests. At least three independent experiments were carried out in triplicate for each assay.

## Figures and Tables

**Figure 1 ijms-22-01888-f001:**
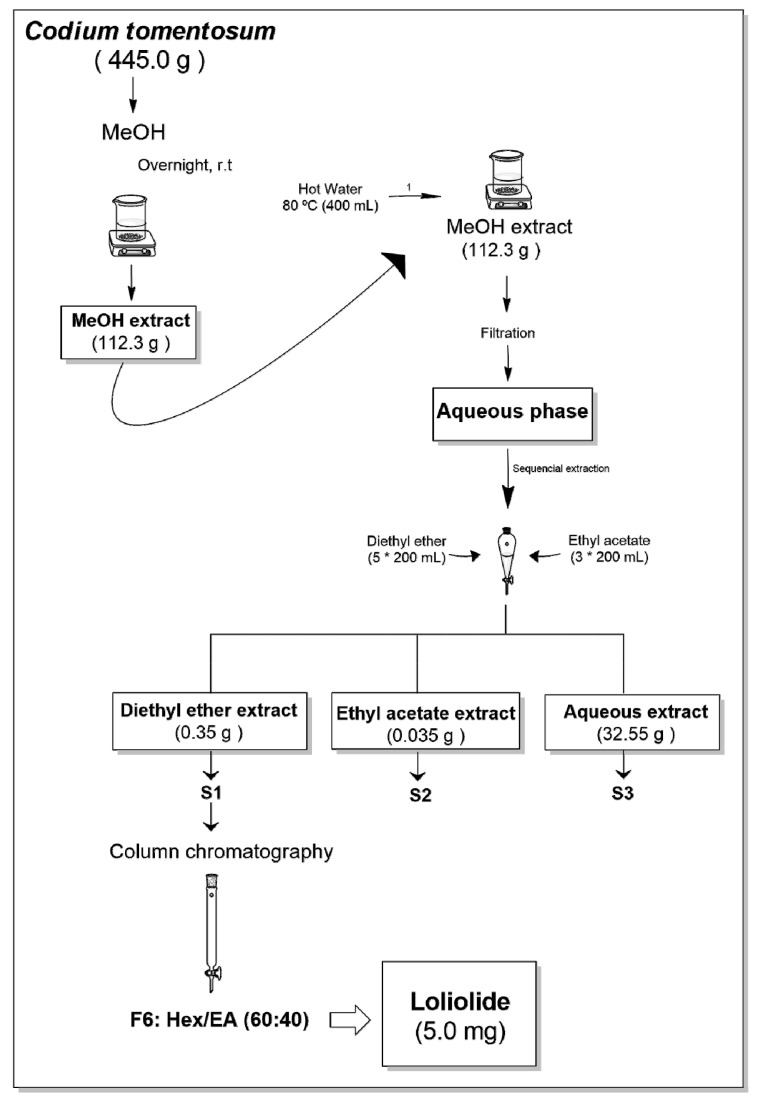
Extraction and fractionation flowchart of the green seaweed *Codium tomentosum*.

**Figure 2 ijms-22-01888-f002:**
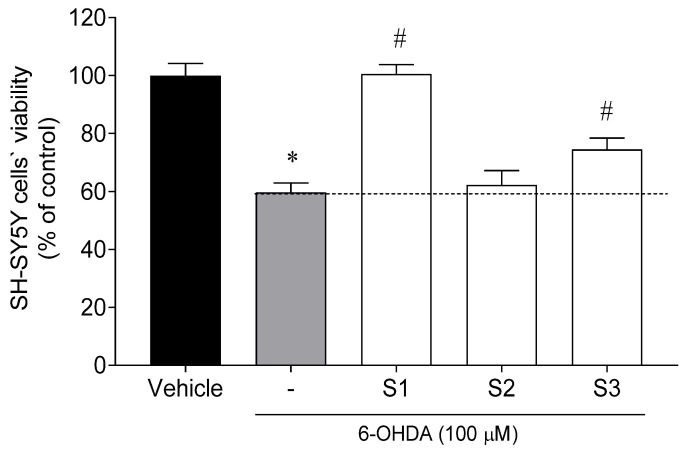
Neuroprotective effects of *Codium tomentosum* fractions (S1–S3 at 100 µg/mL, 24 h) in the presence of 6-OHDA (100 µM) on SH-SY5Y cells. (−) 6-OHDA. The values in each column represent the mean ± standard error of the mean (SEM) of 3 or 4 independent experiments. Symbols represent significant differences (ANOVA, Dunnett’s test, *p* < 0.05) when compared to: * vehicle and ^#^ 6-OHDA.

**Figure 3 ijms-22-01888-f003:**
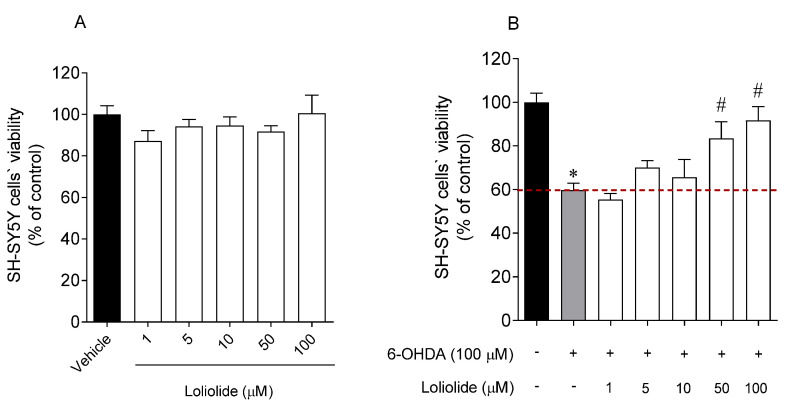
(**A**) SH-SY5Y cells’ viability when exposed 24 h to Loliolide (1–100 µM). (**B**) Neuroprotective effects of Loliolide (1–100 µM) on SH-SY5Y cells exposed to 6-OHDA after 24 h of incubation. (+) With 6-OHDA and (−) without 6-OHDA.The values in each column represent the mean ± standard error of the mean (SEM) of 3 or 4 independent experiments. Symbols represent significant differences (ANOVA, Dunnett’s test, *p <* 0.05) when compared to: * vehicle and ^#^ 6-OHDA.

**Figure 4 ijms-22-01888-f004:**
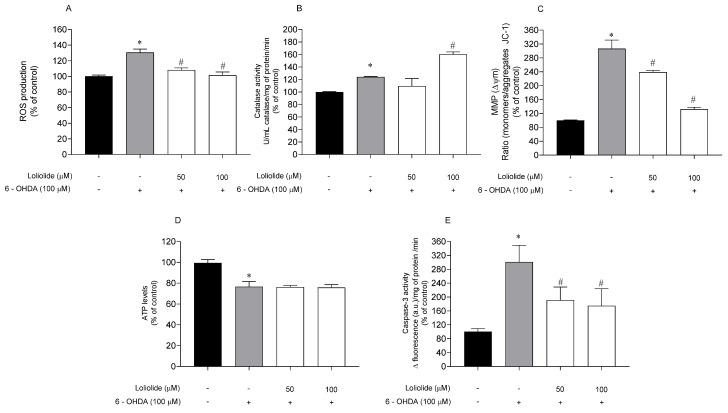
Parkinson’s disease hallmarks associated with neuroprotective effects of Loliolide (50–100 µM, 6 h) determined on SH-SY5Y cells. (**A**) ROS production; (**B**) Catalase activity; (**C**) Mitochondrial membrane potential; (**D**) ATP levels; (**E**) Caspase–3 activity. (+) with 6-OHDA and (−) without 6-OHDA. The values in each column represent the mean ± standard error of the mean (SEM) of 3 or 4 independent experiments. Symbols represent significant differences (ANOVA, Dunnett’s test, *p <* 0.05) when compared to: * vehicle and ^#^ 6-OHDA.

**Figure 5 ijms-22-01888-f005:**
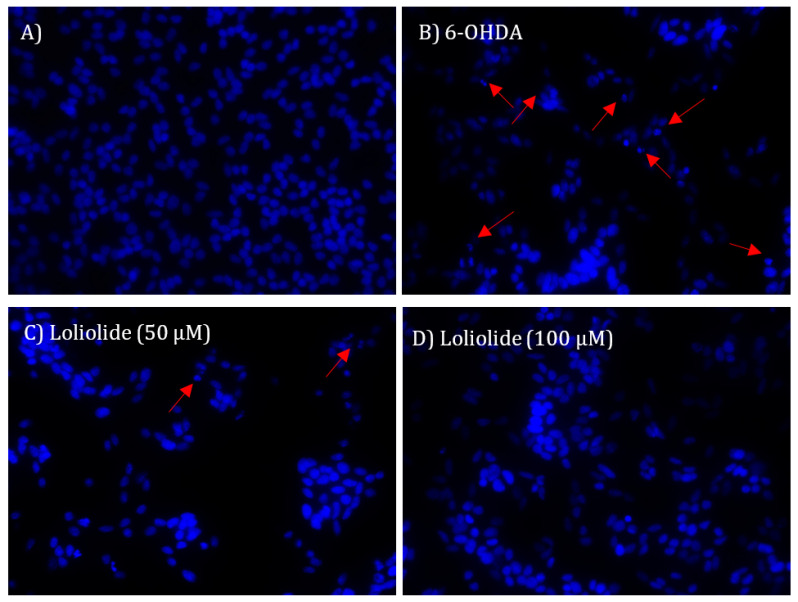
Nuclear morphology of SH-SY5Y cells stained with DAPI probe. (**A**) No treatment; (**B**) Cells exposed to 6-OHDA (100 µM; 24 h); (**C**) Cells exposed concomitantly to Loliolide (50 µM) and 6-OHDA (100 µM) for 24 h; (**D**) cells exposed concomitantly to Loliolide (100 µM) and 6-OHDA (100 µM) for 24 h. The DNA fragmentation pattern is an indicator of apoptosis. Arrows point to nuclear fragmentation. The images are representative of one well of each situation tested.

**Figure 6 ijms-22-01888-f006:**
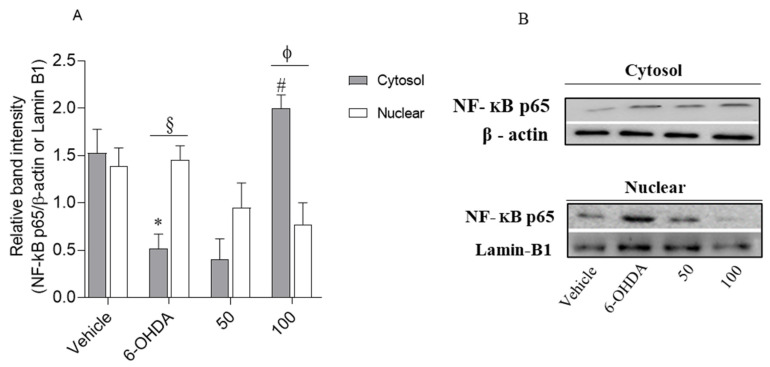
Comparison of the NF-kB p65 expression levels in the cytosol and nucleus in response to 6-OHDA (100 µM), in the absence and presence of Loliolide (50–100 µM). (**A**) The values in each column represent the mean ± standard error of the mean (SEM) of 3 or 4 independent experiments. Symbols represent significant differences (ANOVA, Dunnett’s test, *p <* 0.05) when compared to: * vehicle, and ^#^ 6-OHDA, ^§^ situation of nucleus and ^ϕ^ situation of 50 µM. (**B**) Relative protein expression levels based on Western-blot band intensity; protein levels were normalized with β-actin and Lamin-B1 in the cytosol and nucleus, respectively.

**Figure 7 ijms-22-01888-f007:**
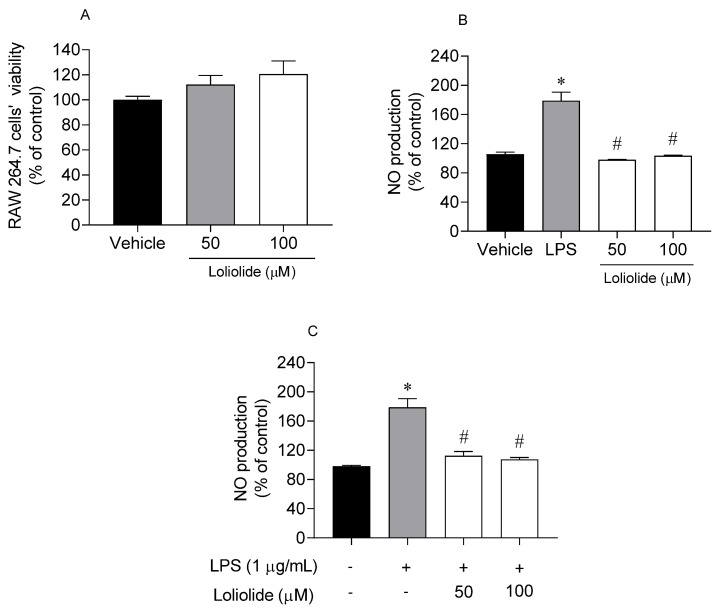
Evaluation of cells’ viability and inflammation on RAW 264.7 cells for 24 h. (**A**) RAW 264.7 cells’ viability exposed to Loliolide (50–10 µM); (**B**) nitric oxide (NO) production by RAW 264.7 cells exposed to Loliolide (50–10 µM); (**C**) NO production by RAW 264.7 cells exposed to lipopolysaccharide (LPS) (1 µg/mL) and Loliolide (50–10 µM). (+) with LPS and (−) without LPS. The values in each column represent the mean ± standard error of the mean (SEM) of 3 or 4 independent experiments. Symbols represent significant differences (ANOVA, Dunnett’s test, *p <* 0.05) when compared to: * vehicle and ^#^ LPS.

**Figure 8 ijms-22-01888-f008:**
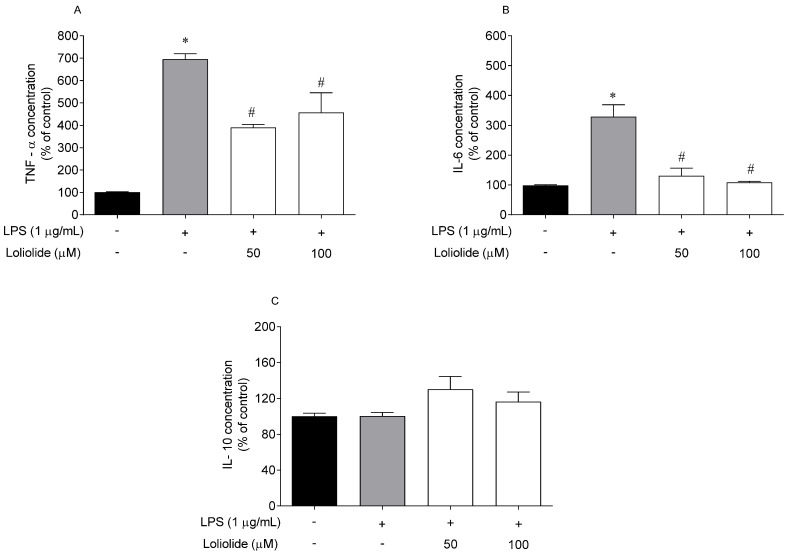
Levels of pro-inflammatory cytokines on RAW 264.7 cells after exposure to Loliolide (50–100 µM) and to LPS (1 µg/mL) for 18 h. (**A**) TNF-α; (**B**) IL-6; (**C**) IL-10. (+) with LPS and (−) without LPS. The values in each column represent the mean ± standard error of the mean (SEM) of 3 or 4 independent experiments. Symbols represent significant differences (ANOVA, Dunnett’s test, *p <* 0.05) when compared to: * vehicle and ^#^ LPS.

**Figure 9 ijms-22-01888-f009:**
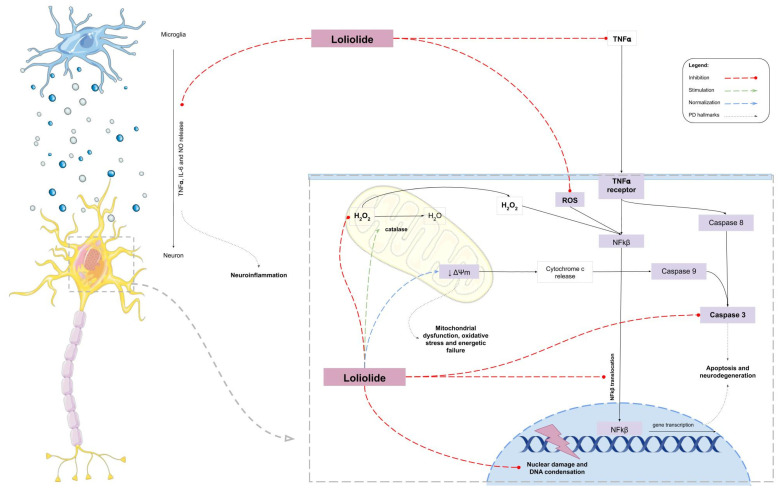
Hypothetical mechanism of action of Loliolide in 6-OHDA-induced cell death and in LPS-induced inflammation.

**Table 1 ijms-22-01888-t001:** Extraction yields and antioxidant activity of *Codium tomentosum* fractions.

Fraction	Yield	DPPH ^a^	FRAP ^b^
S1	0.31%	>200	82.01 ± 0.97
S2	0.03%	>200	8.54 ± 2.66
S3	28.98%	>200	6.07 ± 0.16
BHT	-	143.70 ± 23.26	2821.50 ± 63.03

^a^ radical scavenging activity (EC_50_ µg/mL); ^b^ µM FeSO_4_/g extract; EC_50_ values were determined for a 95% confidence interval.

**Table 2 ijms-22-01888-t002:** Nuclear magnetic resonance (NMR) data (400 MHz, CDCl_3_) of Loliolide isolated from *Codium tomentosum*.

Position	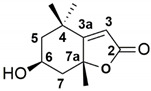 Loliolide (1)	
	*δ* C	*δ*H, m, *J*(Hz)
2	172.07	-
3	112.88	5.69 s
3a	182.60	-
4	35.91	-
5	47.26	1.52 dd, 14.5, 3.2, α-H*_ax_*
		1.98 brd, 14.5, β-H*_eq_*
6	66.79	4.33 m, α-H*_eq_*
7	45.57	1.79 m, α-H*_ax_*
		2.46 brd, 14.2 β-H*_eq_*
7a	86.83	-
4α-Me	30.67	1.27 s, Me*_eq_*
4β-Me	26.48	1.46 s, Me*_ax_*
7a-Me	26.96	1.78 s, β-Me*_ax_*

## Data Availability

The data presented in this study are available on request from the corresponding author.

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
