# Peer review of "Loliolide, a New Therapeutic Option for Neurological Diseases? In Vitro Neuroprotective and Anti-Inflammatory Activities of a Monoterpenoid Lactone Isolated from Codium tomentosum"

_ijms, 2021, doi:10.3390/ijms22041888_

Round 1
Reviewer 1 Report
The manuscript from Silva et al. describes the neuroprotectant properties of Loliolide as the principal component from fractions of the seaweed Codium tomentosum.
The authors carried out the collection, fractioning, purification, and structural elucidation of the most active components, from those fractions that shown some neuroprotectant in vitro activity against SH-SY5Y cells under oxidative stress conditions.
Although the results obtained from the experiments performed agree with the neuroprotectant activity described for Loliolide, however, there are some issues that the authors should address:
- The catalase activity levels detected after treatment with Loliolide 50uM and 100uM are confused. The authors may perform these experiments with other methods, considering that Catalase determinations are highly dependent on the determination assay.
- The same arguments mentioned in the former paragraph applied to the ATP determination levels.
- It could be worth studying the antiapoptotic capacity of Loliolide in the SH-SY5Y cell line under 6-OHD pressure to see if there is any correlation with the Caspase-3 results presented here.
- It would be interesting to study by western blot the protein expression levels of other endogenous antioxidant enzymes (SOD, GSH-Px), and if there is an upregulation effect of Loliolide on the antioxidant system Nerf/OH-1.
Author Response
We sincerely appreciate all the suggestions and promptly complete the manuscript with all the changes recommended in order to answer to theirs concerns, ensuring all conditions to be accepted for publication in International Journal Molecular of Sciences.
- Reviewer Comments
“The manuscript from Silva et al. describes the neuroprotectant properties of Loliolide as the principal component from fractions of the seaweed Codium tomentosum. The authors carried out the collection, fractioning, purification, and structural elucidation of the most active components, from those fractions that shown some neuroprotectant in vitro activity against SH-SY5Y cells under oxidative stress conditions. Although the results obtained from the experiments performed agree with the neuroprotectant activity described for Loliolide, however, there are some issues that the authors should address:
- The catalase activity levels detected after treatment with Loliolide 50uM and 100uM are confused. The authors may perform these experiments with other methods, considering that Catalase determinations are highly dependent on the determination assay.
Answer: In this study loliolide exhibited weak antioxidant activity by the chemical assays (FRAP, DPPH, ORAC), which are directly related with the neutralization of free radicals. However, when tested in the SH-SY5Y cellular model exposed to 6-OHDA neurotoxin, loliolide was able to decrease the ROS levels and mitochondrial membrane potential depolarization induced by 6-OHDA treatment as well as increase Catalase activity. These results suggest that its activity may be not directly related with the neutralization of ROS but with the stimulation of the antioxidant defenses as Catalase activity. This point of view is reinforced by the results attained with loliolide when tested at 100 µM, being possible to observe that the highest increase of Catalase activity was accompanied by the highest decrease of ROS levels (Figures 4A and 4B). Regarding 6-OHDA, it was observed a slight increase of Catalase activity that can be explained by the cellular metabolism response to the increase of ROS levels, such as H2O2, mediated by 6-OHDA treatment, leading to a stimulation of enzyme activity in order to detoxify H2O2. We hope that this explanation may contribute to clarify the results obtained in the Catalase activity. Concerning the suggestion of reviewer, unfortunately, at this moment it will not be possible to accomplish additional experimental assays due to the pandemic situation and the emergence state in Portugal that conditioned the access to laboratory facilities. Accordingly, the Discussion section was improved, and changes highlighted (Lines 395-405; 411-414).
- The same arguments mentioned in the former paragraph applied to the ATP determination levels.
Answer: In fact, it was possible to observe a decrease of mitochondrial dysfunction induced by 6-OHDA when SH-SY5Y were treated in the presence of loliolide. However, this mitochondrial protection mediated by loliolide was not reflected in the recuperation of ATP levels, which maintained similar to 6-OHDA condition. Based in the results attained, we hypothesize that it would be necessary to increase the time of incubation in order to be possible to observe an increase of ATP levels. Nevertheless, further studies should be considered in order to prove this theory. Therefore, the Discussion section was improved, and the changes are highlighted in the manuscript (Lines 426-431). Concerning the reviewer suggestion, unfortunately, it will be not possible to accomplish additional experimental assays due to the reasons above described.
- It could be worth studying the antiapoptotic capacity of Loliolide in the SH-SY5Y cell line under 6-OHD pressure to see if there is any correlation with the Caspase-3 results presented here.
Answer: Previous studies have been reported that both the intrinsic and extrinsic apoptotic pathways may be implicated in the neurodegeneration of dopaminergic neuronal cells in Parkinson’s disease. Despite these pathways can be initiated by distinct initiator caspases, namely -9 and -8, respectively, both converge onto a common pathway of executioner caspases, involving Caspase-3 and Caspase-6. Therefore, Caspase-3 was selected as key biomarker of apoptosis, in order to verify if the neuroprotective ability of lioliolide was or wasn’t related with the blockage of apoptosis. Furthermore, the relevance of this biomarker in the pathogenesis of Parkinson’s disease is reinforced by the previous evidence in postmortem and in vitro studies, in which it was verified an elevated activity and expression of Caspase-3 in substantia nigra pars compacta. However, we fully agree with reviewer that in order to depict the antiapoptotic effects of the compound further studies should be considered to evaluate its effects in other apoptosis biomarkers (e.g. Caspases -9 and -8, Bax, cytochrome C). Thus, the Discussion section was improved, and changes highlighted (Lines 497-507).
- It would be interesting to study by western blot the protein expression levels of other endogenous antioxidant enzymes (SOD, GSH-Px), and if there is an upregulation effect of Loliolide on the antioxidant system Nerf/OH-1.
Answer: Due to the promising activities exhibited by loliolide in this study, we fully agree with reviewer that further studies should be outlined in order to fully depicted the intracellular signaling pathways underlying its neuroprotective activity. Thus, the study of protein expression levels of other endogenous antioxidant enzymes (SOD, GSH-Px) as well as its effects in the upregulation effect of loliolide on the antioxidant system Nrf2/OH-1, as suggested by the reviewer, is of utmost relevance. Therefore, due to the interesting results attained in the present study and considering our limitations at this moment, we decided to improve the Discussion section highlighting the further research directions that should be accomplished in order to fully depict the intracellular signaling pathways underlying the neuroprotective potential of loliolide. The changes are highlighted in the manuscript (Lines 497-507).
Reviewer 2 Report
This study deals with Loliolide, a monoterpenoid lactone extracted from the green seaweed Codium tomentosum, and its in vitro effects on two different cellular models treated with the neurotoxin 6-hydroxydopamine (6-OHDA). Loliolide was able to counteract, at least in part, the 6-OHDA-related toxicity, showing a clear protective action. As the cellular lines treated with 8_OHDA models, may be considered as experimental models of Parkinson's Disease (PD), the authors take their results valid for translation studies of Loliolide efficacy from the cell lines to PD therapy.
This work is the natural prosecution of the previous article: “Natural Approaches for Neurological Disorders-The Neuroprotective Potential of Codium tomentosum” by Silva J, Martins A, Alves C, Pinteus S, Gaspar H, Alfonso A, Pedrosa R.; Molecules. 2020 Nov 23;25(22):5478. doi: 10.3390/molecules25225478.
The work is accompanied by several experiments and numerous data but presents some critical issues to be improved.
1) The 2nd part of the title should be changed. A possible example may be: “In vitro neuroprotective and anti-inflammatory activity of Loliolide, a molecule extracted from the green seawood Codium Tomentosum. A new therapeutic option for Parkinson’s Disease and other neuroinflammatory conditions?”
2) The authors place a lot of emphasis on Loliolide as an antioxidant, but in their evaluation of antioxidant power they do not find corresponding results. Loliolide seems to have a clear anti-inflammatory action and it is this that must be discussed, not the antioxidant action. In this context, the authors should explain the discrepancies between what is reported about the antioxidant power in Table 1 (page 3/23) and paragraph 2.3.1 (page 5/23). Furthermore, the authors should rather focus their exposition on chronic inflammation. There is recent work indicating that chronic neuroinflammatory diseases can originate from intestinal inflammation and can depend on the type of diet. The two factors, diet and gut microbiota, can cooperate, damage the blood-brain barrier and cause diseases such as PD. In this context, Loliolide, as a dietary anti-inflammatory molecule, can represent an excellent means to combat the disease.
3) In general, this work must be clearly presented as a continuation of the previous one (see above), not only as reference 22 hidden between references 21,23 (line 100).
4) Each paragraph in Materials and Methods should possibly have a corresponding short paragraph in the Results.
Some specific comments and/or corrections in detail:
A. Loliolide always with initial capital letter or always lowercase.
B. Always write in full the abbreviations used for the first time in the text (MTT, MMP, etc…).
C. Present the cell lines used when they are considered for the first time.
D. When introducing the transferability of results to Parkinson's disease, mention that Loliolide can pass through the Blood Brain Barrier (BBB).
E. Authors should mention that loliolide can be purified from sources other than Codium Tomentosum, in particular from other algae.
F. Discussion should be shortened and possibly divided in paragraphs. Too much space for catalase stimulation, which does not have its own paragraph in the results. The same is true for mitochondrial dysfunction (from line 367).
G. Consider that the production of ROS, TNF-alfa, and pro-inflammatory interleukins has the same origin: the activation of NF-kB. IL-10 is an anti-inflammatory interleukin.
H. In some parts (lines 201-218; 242-253, 267-274) the readability of the text is difficult. It would be better to organize tables to associate with figures 4,6,7 or to insert the corresponding values in the figures, avoiding writing them in the text.
I. Page 3/23: Table 1 is unclear. The legend needs to explain what S1, S2, S3 are, what are DPPH and FRAP and should refer Table 1 to Figure 1.
J. Page 4/23, line 155: the expression: "fully revert" would be fine if the Loliolide was added after 6-OHDA. However, if both were added at the same time a better expression would be: contrasts, limits, counteracts (or similar) the toxicity of 6-OHDA.
K. Page 5/23, line 169: separate “hasnot”.
L. Page 6/23, line 217: ….in about?
M. Page 7/23, create a paragraph dedicated to the inhibition of DNA fragmentation with its own sub-title and align the legend of figure 5.
N. Page 7/23, line 230/231: NF-kB is not only in the brain.
O. Page 10/23, line 296: in both cases…?
P. Page 10/23, line 304: to disclose…
Q. Page 13/23: Figure 9 is not readable.
R. Page 14/23, Figure 1: write Codium tomentosum instead of Codim tomentosum.
Author Response
We sincerely appreciate all the suggestions and promptly complete the manuscript with all the changes recommended as well as to answer to the reviewers’ concerns, in order to ensure all conditions to be accept for publication in International Journal Molecular of Sciences. The responses to reviewers questions are described in the file attached.
Kindly regards,
Joana Silva
We sincerely appreciate all the suggestions and promptly complete the manuscript with all the changes recommended in order to answer to theirs concerns, ensuring all conditions to be accepted for publication in International Journal Molecular of Sciences.
Response to Reviewer 2Comments
This study deals with Loliolide, a monoterpenoid lactone extracted from the green seaweed Codium tomentosum, and its in vitro effects on two different cellular models treated with the neurotoxin 6-hydroxydopamine (6-OHDA). Loliolide was able to counteract, at least in part, the 6-OHDA-related toxicity, showing a clear protective action. As the cellular lines treated with 6-OHDA models, may be considered as experimental models of Parkinson's Disease (PD), the authors take their results valid for translation studies of Loliolide efficacy from the cell lines to PD therapy.
This work is the natural prosecution of the previous article: “Natural Approaches for Neurological Disorders-The Neuroprotective Potential of Codium tomentosum” by Silva J, Martins A, Alves C, Pinteus S, Gaspar H, Alfonso A, Pedrosa R.; Molecules. 2020 Nov 23;25(22):5478. doi: 10.3390/molecules25225478.
The work is accompanied by several experiments and numerous data but presents some critical issues to be improved.
1)The 2nd part of the title should be changed. A possible example may be: “In vitro neuroprotective and anti-inflammatory activity of Loliolide, a molecule extracted from the green seawood Codium Tomentosum. A new therapeutic option for Parkinson’s Disease and other neuroinflammatory conditions?”
Answer: The title was modifiedas suggested by the reviewer.Please, check if it is better now.The changes were highlighted in the manuscript (line 1 -5).
2)The authors place a lot of emphasis on Loliolide as an antioxidant, but in their evaluation of antioxidant power they do not find corresponding results. Loliolide seems to have a clear anti-inflammatory action and it is this that must be discussed, not the antioxidant action. In this context, the authors should explain the discrepancies between what is reported about the antioxidant power in Table 1 (page 3/23) and paragraph 2.3.1 (page 5/23). Furthermore, the authors should rather focus their exposition on chronic inflammation. There is recent work indicating that chronic neuroinflammatory diseases can originate from intestinal inflammation and can depend on the type of diet. The two factors, diet and gut microbiota, can cooperate, damage the blood-brain barrier and cause diseases such as PD. In this context, Loliolide, as a dietary anti-inflammatory molecule,can represent an excellent means to combat the disease.
Regarding the question raised by the reviewer about the discrepancies between what is reported about the antioxidant power in Table 1 (page 3/23) and paragraph 2.3.1 (page 5/23), if we understand correctly the concern raised, it is important to have in attention that correspond to different results. The results in Table 1 correspond to the antioxidant capacity of fractions. The fraction S1 exhibited the highest activity. On the other hand, the paragraph 2.3.1 (page 5/23) corresponds to the antioxidant activity of Loliolide, which was isolated from fraction S1, but presented low potential when
compared with the respectivefraction, suggesting that the capacity observed in thisfractionmay be related with presence of other compounds. We would like to thank reviewer by the suggestion concerning the intestinal inflammation and development of neuroinflammatory diseases that itisan excellent point of view that contributesto improve the qualityof manuscript discussion,as well as to open new research windows.
The Results (Lines175-176) and Discussion (Lines360-365; 475-493) sections were improved as suggested by the reviewer. The changes were highlighted in the manuscript.
3)In general, this work must be clearly presented as a continuation of the previous one (see above), not only as reference 22 hidden between references 21,23 (line 100).
Answer:The Introduction (Lines111-117)section was improved as suggested bythe reviewer. The changes were highlighted in the manuscript.
4)Each paragraph in Materials and Methods should possibly have a corresponding short paragraph in the Results.
Answer:We are not sure if we correctly understood the reviewer concern. However, we try to adjust the Results titles according to the titles of the Materials and Methods. However, we would like to clarify that it is not possible to establish a directly connection because,for instance in the Materials and Methods section the protocol to evaluate the antioxidant potential was used to evaluate the antioxidant activity of seaweed fractions as well as the Loliolide compound. Therefore, the method was used to obtain two “distinct results” that were described in different sub-titles of Results section according to theperformed bioguidedscreening. Finally, if we did not answer correctly to the reviewer concern, we kindly askthe reviewer to reformulate the commentary in order to achieve the reviewer ́request.
Some specific comments and/or corrections in detail:
A.Loliolide always with initial capital letter or always lowercase.
Answer:The Loliolide word was uniformized in the manuscript. The changes were highlighted in the manuscript.
B.Always write in full the abbreviations used for the first time in the text (MTT, MMP, etc...).
Answer:The abbreviations were full writtenwhen they appear for the first time,as suggested.The changes were highlighted in the manuscript (Lines 181-182, 199, 237-238 and 316)
C.Present the cell lines used when they are considered for the first time.
Answer:The manuscript was improved. The changes were highlighted in the manuscript (Lines 180and 314)
D.When introducing the transferability of results to Parkinson's disease, mention that Loliolide can pass through the Blood Brain Barrier (BBB).
Answer:The ability of Loliolideto cross BBB was added in the Discussion sectionas suggested by the reviewer. The changes were highlighted in the manuscript(Lines478-493).
E. Authors should mention that loliolidecan be purified from sources other than Codium Tomentosum, in particular from other algae.
Answer:The introduction section was improved accordingto reviewersuggestion. The changes were highlighted in the manuscript(Lines121-123).
F. Discussion should be shortened and possibly divided in paragraphs. Too much space for catalase stimulation, which does not have its own paragraph in the results. The same is true for mitochondrial dysfunction (from line 367).
Answer: The Discussion section was adjusted according to the suggestions of both reviewers. However, as suggested by the reviewer, some parts of the Discussion were removed/ adapted. We are not sure about what the reviewer intended with the sentence”possibility divided in paragraphs”, thus we decided to divide Discussion section in two different topics, antioxidant and neuroprotective activity of fractions and Loliolide and anti-inflammatory activities of Loliolide. We kindly ask reviewer to check if the performed modifications are in accordance with the requested changes. The changes were highlighted in the manuscript.
G. Consider that the production of ROS, TNF-alfa, and pro-inflammatory interleukins has the same origin: the activation of NF-kB. IL-10 is an anti-inflammatory interleukin.
Answer:The Discussion section was improved. The changes were highlighted in the manuscript (Lines443-446).
H. In some parts (lines 201-218; 242-253, 267-274) the readability of the text is difficult. It would be better to organize tables to associate with figures 4,6,7 or to insert the corresponding values in the figures, avoiding writing them in the text.
Answer:According to reviewer comments as well as the fact that the interpretation of figures and text is not influenced by the presentation of the respective values,we decided to removethem. However, if the reviewer think that it is essential this information we can add it in the figures.
I. Page 3/23: Table 1 is unclear. The legend needs to explain what S1, S2, S3 are, what are DPPH and FRAP and should refer Table 1 to Figure 1.
Answer:The Results (line 131-158) section was improved as suggested by the reviewer. The changes were highlighted in the manuscript, being added the topic “2.1.1. Extraction and fractionation of Codium tomentosumseaweed”that explain the origin of S1-S3 fractions.
J. Page 4/23, line 155: the expression: "fully revert" would be fine if the Loliolidewas added after 6-OHDA. However, if both were added at the same time a better expression would be: contrasts, limits, counteracts (or similar) the toxicity of 6-OHDA.
Answer:It was corrected.The changes were highlighted in the manuscript line (Line 196).
K. Page 5/23, line 169: separate “hasnot”.
Answer:It was corrected (Line 211).
L. Page 6/23, line 217: ....in about?
Answer:It was corrected (Lines254-266).
M. Page 7/23,create a paragraph dedicated to the inhibition of DNA fragmentation with its own sub-title and align the legend of figure 5.
Answer:It was created a paragraph.The changes were highlighted in the manuscript(Lines269-274)
N. Page 7/23, line 230/231: NF-kB is not only in the brain.Answer:The sentence was reformulated. The changes were highlighted in the manuscript (Lines288-289)
O. Page 10/23, line 296: in both cases...?
Answer:The sentence was reformulated. Thechanges were highlighted in the manuscript (Lines355-356)
P. Page 10/23, line 304: to disclose...
Answer:It was corrected. The changes were highlighted in the manuscript (Line 358)
Q. Page 13/23: Figure 9 is not readable.
Answer:The Figure was improved and added tothe manuscript. Please,check if it isreadable.
R. Page 14/23, Figure 1: write Codium tomentosum instead of Codim tomentosum.
Answer:It was corrected. The changes were highlighted in the manuscript (Line 132)